# A single-cell and spatial RNA-seq database for Alzheimer's disease (ssREAD)

Cankun Wang [1], Diana Acosta[2], Megan McNutt[1], Jiang Bian [3], Anjun Ma [1], Hongjun Fu [2,4] ✉ & Qin Ma [1] ✉

Alzheimer's Disease (AD) pathology has been increasingly explored through single-cell and single-nucleus RNA-sequencing (scRNA-seq & snRNA-seq) and spatial transcriptomics (ST). However, the surge in data demands a comprehensive, user-friendly repository. Addressing this, we introduce a single-cell and spatial RNA-seq database for Alzheimer's disease (ssREAD). It offers a broader spectrum of AD-related datasets, an optimized analytical pipeline, and improved usability. The database encompasses 1,053 samples (277 integrated datasets) from 67 AD-related scRNA-seq & snRNA-seq studies, totaling 7,332,202 cells. Additionally, it archives 381 ST datasets from 18 human and mouse brain studies. Each dataset is annotated with details such as species, gender, brain region, disease/control status, age, and AD Braak stages. ssREAD also provides an analysis suite for cell clustering, identification of differentially expressed and spatially variable genes, cell-type-specific marker genes and regulons, and spot deconvolution for integrative analysis. ssREAD is freely available at https://bmblx.bmi.osumc.edu/ssread/.

Alzheimer's disease (AD) is a progressive neurodegenerative disorder, the most common form of dementia worldwide. Over 57 million individuals globally suffer from this debilitating condition[1]. Despite the significant strides in medical research and development, therapeutic interventions for AD remain distressingly ineffective. This glaring lacuna in clinical therapeutics underscores the urgency to probe into the intricate molecular mechanisms underlying the disease's cellular and regional susceptibility that are still largely enigmatic[2]. Recent high-throughput sequencing technologies, particularly single-cell RNA-sequencing (scRNA-seq) and single-nucleus RNA-sequencing (snRNA-seq) have cast fresh light on our exploration of AD pathogenesis. To study the cellular heterogeneity of the brain and reveal the complex cellular changes in AD, we launched scREAD in 2020[3]. By then, it was the first database dedicated to managing public AD-related sc/snRNA-Seq data from human and mouse brain tissue. As the sequencing technology and scientific inquiry rapidly evolved, more studies are discerning the spatial information of differentially expressed genes (DEGs) associated with AD pathology, the interconnectedness of DEGs

related to AD biomarkers, DEGs enriched in specific cell subtypes, cell−cell communications, and regional and cellular vulnerability in AD[4]. Spatial transcriptomics (ST) revolutionized our understanding of neurobiology and AD pathogenesis by enabling the visualization of gene expression patterns within their spatial context. Yet, these public-available scRNA-seq, snRNA-seq, and ST data have not been well collected and managed by any AD databases. In addition, the remarkable potential usage of these datasets is accompanied by the formidable challenge of data aggregation, analysis, and interpretation, necessitating substantial computational resources and bioinformatics expertize.

In recent years, there has been a significant increase in the availability of AD-related sc/sn RNA-seq datasets. This has led to the development of various databases, including those that serve as general repositories for scRNA-seq data and others that specifically focus on AD and the brain. For instance, the TACA[5] database facilitates differential expression comparisons to identify cell type-specific gene expression alterations, cell−cell interactions, and drug screening

[1]Department of Biomedical Informatics, The Ohio State University, Columbus, OH 43210, USA. [2]Department of Neuroscience, The Ohio State University, Columbus, OH 43210, USA. [3]Department of Health Outcomes & Biomedical Informatics, University of Florida, Gainesville, FL 32606, USA. [4]Chronic Brain Injury Program, The Ohio State University, Columbus, OH 43210, USA. ✉e-mail: hongjun.fu@osumc.edu; qin.ma@osumc.edu

opportunities. Meanwhile, the SC2Disease database[6] aims to offer a comprehensive and accurate resource of gene expression profiles across various cell types for 25 diseases. However, the AD data provided by SC2Disease is limited to a single dataset from one brain region (prefrontal cortex). SCAD-Brain[7] is another public database dedicated to AD, with a focus on both human and mouse brain data. It provides more extensive analysis results, such as cell communication analysis and trajectory information. Despite these advances, there remains a gap in the field for a specialized database that concentrates on spatial transcriptomics in AD and offers comprehensive differential analyses under various conditions, such as sex-specific, region-specific, and comparisons between AD and control groups. Integrating these diverse datasets and conditions would prove invaluable for researchers studying the complex landscape of AD.

To address these burgeoning complexities and to meet the scientific community's growing demand for comprehensive, integrated, and accessible data analysis, we introduce **ssREAD** (Single-cell and Spatial RNA-seq databasE for Alzheimer's Disease). It includes 381 ST and 277 sc/snRNA-seq AD-related datasets. These sequencing data enable researchers to investigate transcriptomic alterations in AD compared to the control and their regulatory mechanisms at various resolutions: sub-cellular, cellular, and spatial levels[8,9], which will help uncover the pathogenesis of AD. We also highlight the sc/snRNA-seq and ST data analysis framework in ssREAD, including cell clustering and annotation, DEGs and spatially variable gene identification, cell-type-specific regulon inference, cell–cell communication analysis and functional enrichment analysis. Moreover, the integrative exploration of ST and sc/snRNA-seq data revealed nuanced molecular landscapes that underlie AD, emphasizing ssREAD's unparalleled capability. Beyond that, ssREAD also delivers marked improvements to the user interface. These modifications, grounded in user-centric design principles, advance visibility and usability, fostering an environment conducive to intuitive data visualization and streamlined querying.

## Results

### Overview of ssREAD

ssREAD comprises 381 ST samples from 16 AD-related studies and 1,053 sc/snRNA-seq samples from 85 studies (Supplementary Data 1). The 1,053 sc/snRNA-seq samples are grouped into 277 datasets by integrating sample replications from the same study. All datasets are collected and downloaded from Broad Institute SingleCellPortal, Gene Expression Omnibus (GEO), and Synapse[10] (Supplementary Fig. 1). ssREAD has a considerable 379% increase in the number of integrated sc/snRNA-seq datasets than the previous scREAD (from 73 to 277), encompassing over three times the total cell and nucleus count.

In our pursuit of thoroughness, each dataset is meticulously annotated, providing pertinent details such as species, gender, brain region, disease/control distinction, and AD Braak stages. In the realm of species, 144 out of 277 integrated sc/snRNA-seq datasets are from human samples, whereas 133 datasets are from mouse samples. In contrast, mouse datasets are prominently represented with 319 ST datasets, four times more than human ST datasets (Fig. 1a). When demarcating datasets based on the AD condition, 154 integrated sc/snRNA-seq datasets are assigned to the AD group, as opposed to the

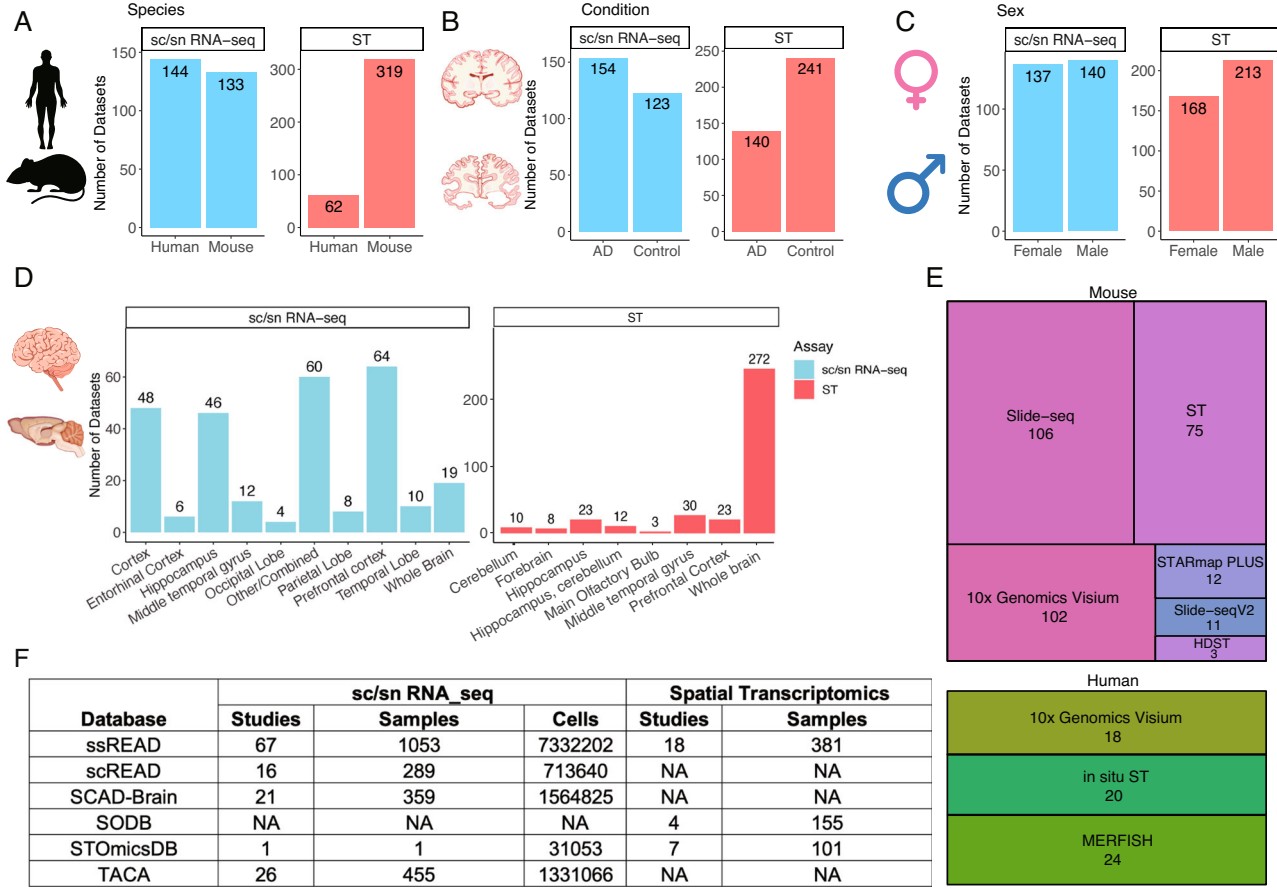

**Fig. 1 | ssREAD Data characteristics and statistics (as of December 2023).** **a–d** Barplots show the number of datasets by species, condition, sex, and brain regions, respectively. **e** Treemap presents the breakdown of technologies deployed in mouse and human studies. **f** The comparative table showcases the number of studies and datasets across extant AD databases and data collection sources.

123 datasets associated with control scenarios (Fig. 1b). This distribution takes a nuanced shift within ST samples, with AD-oriented datasets numbering 140 and control datasets as 241. Sex-specific stratification is also evident within the ssREAD construct. It includes 140 integrated datasets from sc/snRNA-seq and an additional 213 datasets sourced from ST methodologies for males (Fig. 1c), compared to 137 sc/snRNA-seq and 168 ST datasets for females. ssREAD also offers a detailed breakdown of samples by brain region, with separate datasets for sc/snRNA-seq and ST techniques (Fig. 1d). ssREAD showcases a range of technologies that contribute to a comprehensive spatial transcriptomics profile (Fig. 1e). For the mouse model, technologies span from Slide-seq (106 samples) to HDST (3 samples). The human segment is inclined towards 10x Visium, accounting for 18 samples, closely trailed by In-situ ST of 20 samples, and MERFISH with 24 samples.

Benchmarking ssREAD against contemporaneous repositories underscores its unparalleled stature. Compared with some existing databases with a collection of datasets on AD, such as scREAD[3], SCAD-Brain[7], TACA[5], SODB[11], STOmicsDB[12], and Single-Cell Portal, ssREAD has the most decadent AD-related samples collected for both sc/snRNA-seq and ST data with great depth and breadth (Fig. 1f). Overall, ssREAD leads to a comprehensive collection of AD-related scRNA-seq, snRNA-seq, and ST datasets, offering both breadth and depth of data that stand unrivaled compared to other available resources or databases.

Our ssREAD collects the above data and offers comprehensive, in-depth data analyses and result interpretations. For sc/snRNA-seq data, ssREAD provides functions including cell clustering, cell type annotation, marker gene expression visualization, and cell proportion analysis (Fig. 2a). For ST data, it provides visualizations for original spatial H&E image, layer/tissue architecture/spatial domain annotation, marker gene expression on spatial map, and spot deconvolution (Fig. 2b). DEGs can be identified for cell types in sc/snRNA-seq data or spatial layers in ST data (Fig. 2c). Cross-data queries and analyses are also facilitated, including comparative studies between male and female subjects across various datasets. Additionally, spatially variable genes (SVGs) can be identified via spaGCN[13] from ST data to show marker genes with spatially resolved expression patterns that may be related to tissue functions. Functional enrichment analysis is included to identify pathways or gene ontology enriched by DEGs or SVGs (Fig. 2d). Moreover, ssREAD also features cell-type-specific (or layer-specific) regulons for individual datasets and the integrated cell atlas (Fig. 2e), focusing on cellular and regional vulnerability in AD.

To ensure ease of use, all analytical results are displayed via a user-friendly and single-access web portal that frees AD researchers from the requirement of extensive programming knowledge. We offer interactive plots for visualizing cells and spatial spots, including scatter plots, bar plots, and violin plots, as well as real-time analyses for DEGs, SVGs, and functional enrichment queries. Furthermore, all datasets, including author-provided metadata and cell type labels, are provided in ready-to-analyze formats (e.g.,.h5ad and.h5seurat), compatible with analysis tools such as Seurat and Squidpy for further analysis.

## Spatially-informed subpopulation analysis reveals cellular heterogeneity in AD

To illustrate the ST data analysis workflow and functions that are available to users of ssREAD, we used two ST data (ST01101 and ST01103), labeled by six cortical layers and the adjacent white matter in two human middle temporal gyrus (MTG) brain samples based on the information provided in the original study[14] (Fig. 3a). We performed DEG analysis between AD (ST01103) and control (ST01101) for each functional layer (Supplementary Data 2) and evaluated the DEG consistency with the other two comparison groups (i.e., ST01102 vs ST01104 and ST01106 vs ST01105) (Supplementary Data 3-4). Results showed that there exist DEGs between AD and control in each layer that overlapped among the three comparison groups (Supplementary

Fig. 2). The overlapping DEGs among the three comparison groups correspond to up-regulated genes in AD compared to controls of layer 4 with three DEGs, layer 5 with two DEGs, and layer 6 with three DEGs. Further implementation of RESEPT[15], a deep-learning framework for spatial domain detection, provided a precise delineation of the tissue architecture and functional zones in both control and AD brain tissues (Fig. 3b). The resulting spatial delineation remains comparable to the six cortical layers identified in the original study. Layers 5-6 of the AD sample (ST01103) exhibit slight differences in their delineation compared to controls and their original labels, suggesting there may be underlying differences in the functional zones of these layers. The data is consistent with previous publications suggesting layer 5 is highly relevant to changes in AD including accumulation of neurofibrillary tau tangles[16–18]. Moreover, the potential of ssREAD in navigating the complex spatial information of AD was further exemplified through a multi-dimensional exploration of spatially informed sub-populations via MAPLE[19] (Fig. 3c). Noted that MAPLE has a critical multi-sample design considering information sharing across samples and accommodating spatial correlations in gene expression patterns within samples. Thus, clusters MAPLE identifies are sample-specific and could be either shared among samples (e.g., cluster 1 in both ST01101 and ST01103) or unique in individual samples (e.g., cluster 5 only in ST01103). This analysis demonstrated the ability of ssREAD to untangle the spatial complexity inherent in the transcriptomic landscape of AD. To further illuminate these cellular dynamics, we mapped the MAPLE cluster annotations in individual samples to their corresponding layer annotations (Fig. 3d, Supplementary Fig. 3, and Source Data 1). In doing so, we show that cluster 1 corresponds to layers 5 and 6 of the AD sample (ST01103), but only corresponds to layer 6 of the control sample (ST01101). This indicates the deviations in layer 5 of AD cases compared to controls may also be a result of differences in their spatial organization as well as their gene expression patterns.

At the molecular level, we identified DEGs in each MAPLE cluster (Fig. 3e and Source Data 2) and between control and AD samples within individual MAPLE clusters (Fig. 3f, Supplementary Data 5, and Source Data 3). The expression pattern of these DEGs underscores the molecular heterogeneity in MAPLE clusters (subpopulations) and between AD and the control (Fig. 3f). For example, the expression of DEPP1, also known as PGC-1α (peroxisome proliferator-activated receptor gamma coactivator 1-alpha), was found to be significantly lower in AD than the control in our dataset. The PGC-1α is highly responsive to numerous forms of environmental stress, including temperature and nutritional status. Several studies have reported that the level of PGC-1α significantly decreases in the brains with AD compared to control brains[20–23]. PGC-1α has thus been suggested to contribute to the improvement of AD pathophysiology. Additionally, the pathway enrichment analysis on AD and control DEGs in cluster 1 showed both upregulated and downregulated pathways. Interestingly, all five downregulated pathways are associated with immune responses/functions (Fig. 3g), which may highlight the important role of immune cells in AD pathogenesis[24–27]. We then further investigated the association of genes within each pathway to microglial states previously published by Sun et al.[28], as well as disease-associated microglia (DAM) and activated response microglia (ARM) genes[29,30]. In agreement with previously published data, three out of five of our upregulated pathways (Regulation of Expression of Slits and Robos, Selenoamino Acid Metabolism, and Eukaryotic Translation Elongation) include several genes that make up a population of microglia (MG3) that is highly enriched with disease-associated microglial genes. On the other hand, the five downregulated pathways (Scavenging of Heme from Plasma, Interleukin 10 Signaling, Tnfs Bind their Physiological Receptors, Fcgr Activation, and Creation of C4 and C2 Activators) are composed of genes found in microglia states that are associated with inflammation due to presence of cytokine and cytokine receptor-related genes (Supplementary Fig. 4). Therefore, the upregulated pathways overlap

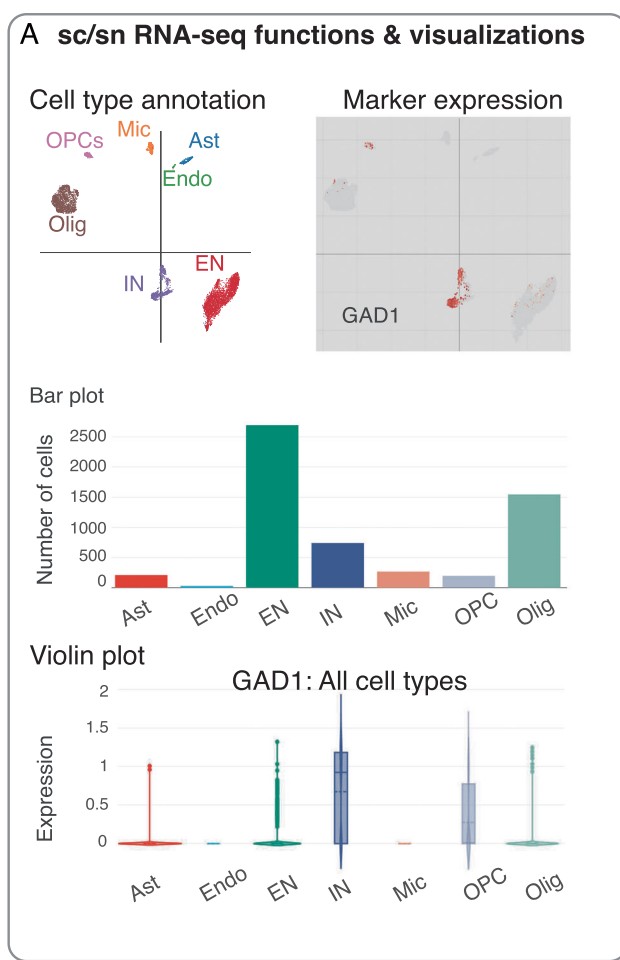

**A  sc/sn RNA-seq functions & visualizations**

Cell type annotation

Marker expression

GAD1

Bar plot

Violin plot

GAD1: All cell types

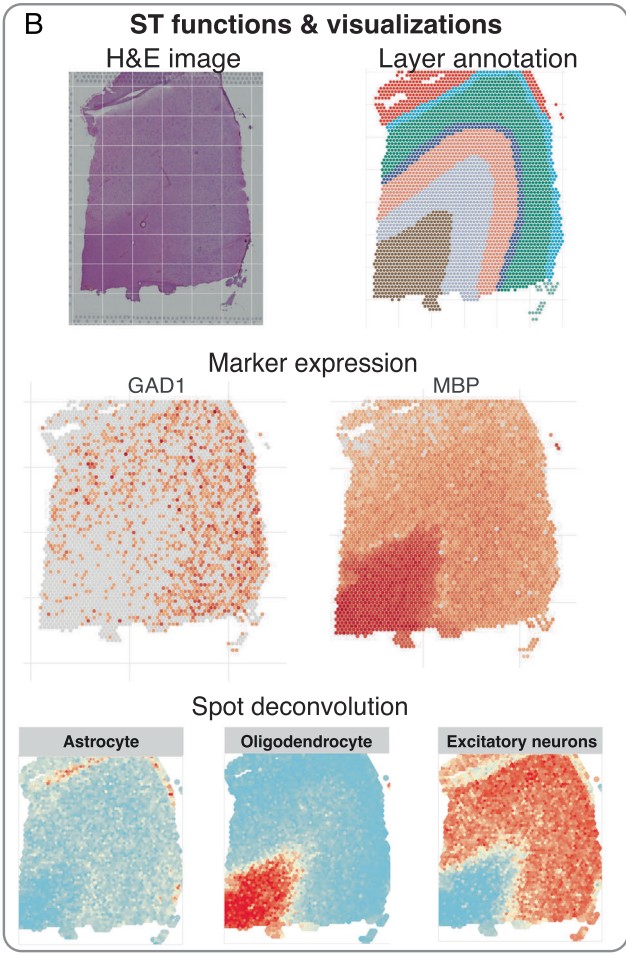

**B  ST functions & visualizations**

H&E image

Layer annotation

Marker expression

GAD1                    MBP

Spot deconvolution

Astrocyte    Oligodendrocyte    Excitatory neurons

**C  Differential gene expression**

| Gene | logFC | Adj. p-val | Pct.1 | Pct.2 |
|------|-------|-----------|-------|-------|
| GAD1 | 1.61 | 9.1E-44 | 0.947 | 0.354 |
| MBP | -0.92 | 3.9E-09 | 0.195 | 0.432 |
| ID2 | -0.67 | 0.0000094 | 0.227 | 9.428 |
| CD163 | 1.41 | 1.1E-30 | 0.232 | 0.092 |

| CT1 vs CT2 | AD vs control |
|---|---|
| Male vs Female | Region 1 vs Region 2 |
| Late vs Early stage | Others comparisons |

**D  Functional enrichment analysis**

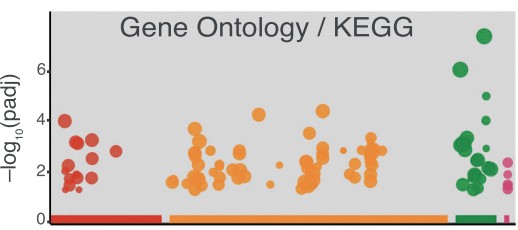

Gene Ontology / KEGG

**E  Cell-type-specific regulons**

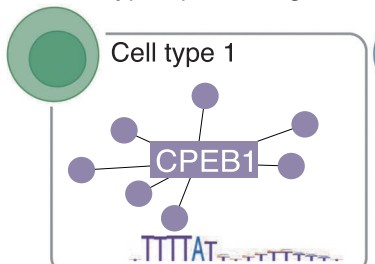

Cell type 1

CPEB1

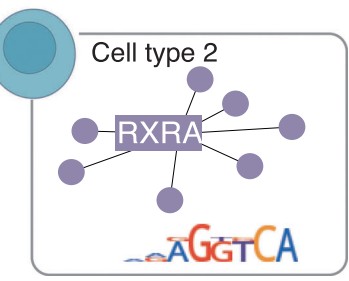

Cell type 2

RXRA

**Fig. 2 | Overview of ssREAD functions. a** Features and visual representations related to sc/snRNA-seq, encompassing cell type annotations, marker gene expressions, and graphic depictions via bar and violin plots. Each box showcases the minimum, first quartile, median, third quartile, and maximum average expression values of a cell type (Ast: *n* = 208, Endo: *n* = 28, EN: *n* = 2,696, IN: *n* = 740, MIC: *n* = 266, OPC: *n* = 195, and Olig: *n* = 1,546). Dots represent outliers. **b** Functions and visualizations pertinent to ST, highlighting H&E imagery, layer annotations, marker gene expressions, and spot deconvolutions. **c** DEG analysis, with comparisons drawn between categories like Cell Type 1 vs. Cell Type 2 (CT1 vs. CT2), AD vs.

Control, Male vs. Female, Brain Regions 1 vs. 2, etc. *p*-values were calculated based on two-sided Wilcoxon Rank–sum test and adjusted using Bonferroni correction. **d** Functional enrichment analysis focusing on GO ontology and KEGG pathways. *p*-values were calculated using the Hypergeometric test from Enrichr and were adjusted using the Benjamini–Hochberg correction method. **e** Predictions of cell type-specific regulons. The following abbreviations are used for cell types: *Ast* astrocytes, *Endo* endothelial cells, *EN* excitatory neurons, *IN* inhibitory neurons, *Mic* microglia, *Olig* Oligodendrocytes, *OPC* oligodendrocyte precursor cells.

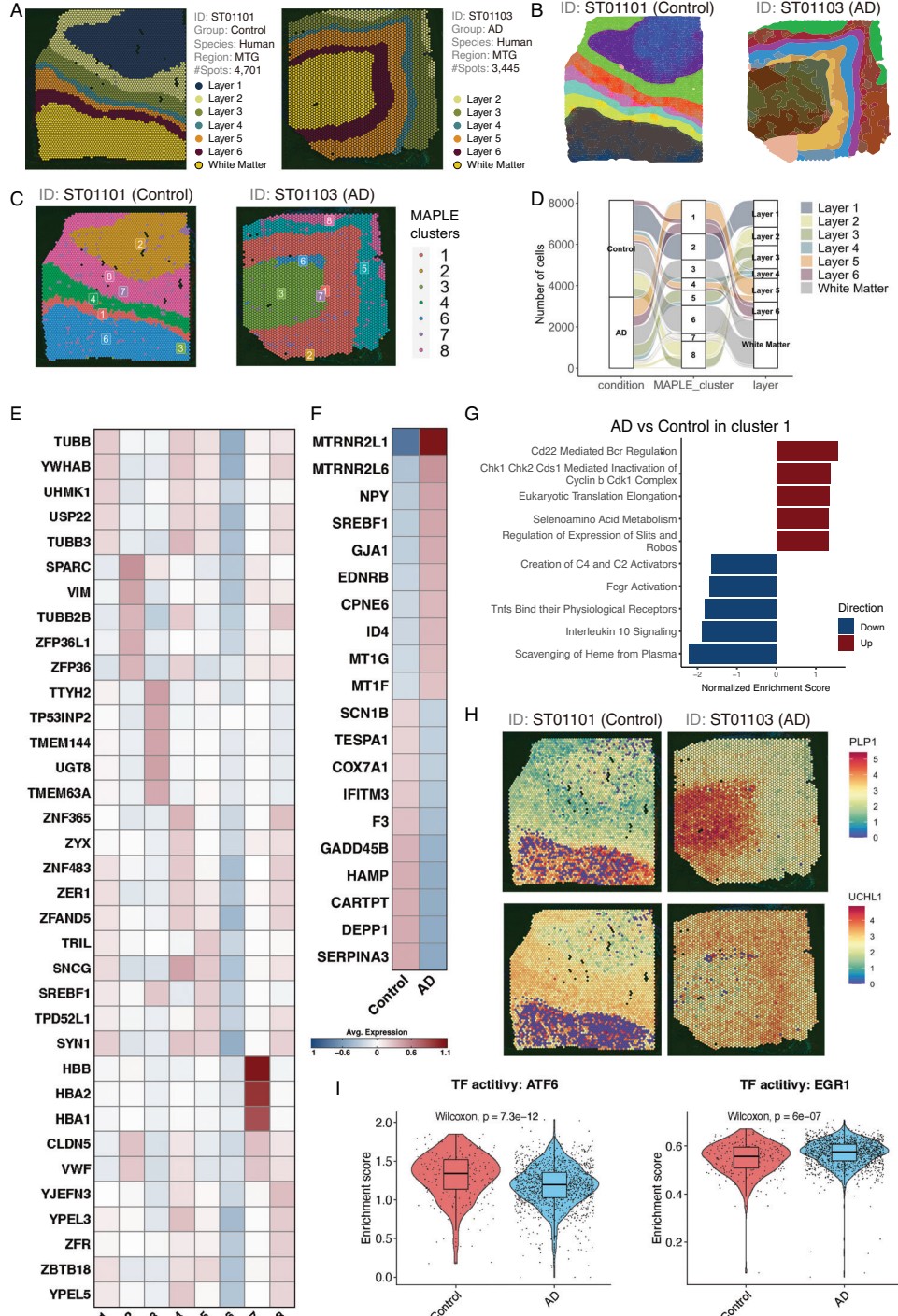

**Fig. 3 | Multi-dimensional analysis of spatially-informed sub-populations.**
**a** Annotation of the six cortical layers alongside the adjacent white matter within two human middle temporal gyrus (MTG) brain samples (ST01101 and ST01103). **b** Detection of spatial domains by RESEPT. **c** Visualization using MAPLE to elucidate shared or unique spatial domains identified across the two Spatial Transcriptomics samples (ST01101 and ST01103). **d** Alluvial diagrams showcasing the progression of cells: originating from individual samples, aggregating into joint subpopulations, and culminating in layer annotations. **e** A heatmap depicting genes specific to the MAPLE-derived clusters for both AD and Control samples. **f** Heatmap representing the top 10 upregulated and top 10 downregulated genes distinguishing AD from Control within Cluster 1. **g** Gene Set Enrichment Analysis (GSEA) of DEGs from

(F) plotted against REACTOME pathways. The bar plot shows the top 10 upregulated and downregulated pathways, accompanied by normalized enrichment scores. **h** Spatial feature plots highlighting the variance in gene expression of *PLP1* and *UCHL1* from Cluster 1, segregated by AD and Control samples. **i** Violin plots showcasing the activity of two selected TFs between AD and Control, with associated *p*-values calculated from a two-sided Wilcoxon rank-sum test. Each box showcases the minimum, first quartile, median, third quartile, and maximum ARI results of a tool performed on different data subsets (Control group: n = 232, and AD group: n = 1412). Dots represent spatial spots. Source data are provided as a Source Data file.

with disease-associated microglia states while downregulated pathways do not include genes found in disease-associated microglia states. In line with previous findings, inflammatory states of microglia change in relation to AD progression, with more strongly correlated inflammatory states being present at early disease stages and weaker inflammatory states present at late disease stages. Therefore, the downregulation of inflammatory-related states in our data coincides with the dysregulation of microglia states found in AD. Besides DEGs and DEG-enriched pathways, we also identified 305 SVGs in cluster 5, such as *PLP1* and *UCHL1* (Fig. 3h), which show clear spatial expression patterns that are linked to specific tissue layers. Interestingly, both genes have also been found to be associated with AD[31,32].

To further investigate the regulatory mechanisms in different clusters between the above Control (ST01101) and AD (ST01103) samples, we implement DeepMAPS[33] in the ssREAD framework for spatial transcriptomics-guided gene regulatory network analysis. This facilitated a comprehensive investigation into the interconnected relationships among key transcription factors (TFs) and their associated genes. Our analysis highlighted a network comprising 10 TFs (i.e., *NR1H3*, *SREBF1*, *ATF6*, *TAL1*, *SOX10*, *NFYA*, *AR*, *MYC*, *HIF1A*, and *EGR1*) and 2,164 genes regulated by those TFs in MAPLE cluster 1 (Supplementary Data 6). This network, interconnected with the enriched genes from gene modules identified by DeepMAPS, underscores the dynamic and intricate interactions between critical TFs and their downstream regulated genes. Further dissection of these TFs showed differences in TF activities between control and AD samples, for example, *ATF6* and *EGR1* (Fig. 3i). These two regulators are reported to be associated with AD pathology and highly related to stress response. *ATF6*, a key player in unfolded protein response to endoplasmic reticulum (ER) stress, has been found to reduce amyloid-beta (Aβ) toxicity via the downregulation of β-site APP-cleaving enzyme 1 (BACE1)[34]. On the other hand, *EGR1* may impair the brain's cholinergic function in the preclinical stages of AD via the upregulation of acetylcholinesterase (AChE)[35], and *EGR1* regulates tau phosphorylation and Aβ synthesis in the brain by enhancing activities of Cdk5 and BACE-1, respectively[36]. The GO and REACTOME enrichment analysis showed that genes regulated by these two TFs are enriched in multiple stress-related pathways. Our results showed that genes, including *XBP1*, *HSPA5*, *DDIT3*, *SEL1L*, *ATP2A2*, and *HSP90B1*, regulated by *ATF6* are related to response to ER stress pathways (Supplementary Data 7). While 106 genes regulated by *EGR1* are related to stress response, such as *CDKN1A*, *MYC*, *TP53*, and *RXRA*, responding to pathways including oxidative stress, ER stress, chemical stress, and stress-activated MAPK cascade. Overall, our data-driven analysis of ST data provides a stepping stone for future studies aimed at deciphering the complex molecular pathogenesis and novel therapeutic targets.

## ssREAD unveils AD pathophysiology through an integrated analysis of spatial and single-cell transcriptomics

One of the most prominent analyses that ssREAD can power is the spot deconvolution enabled by the integration of sc/snRNA-seq and ST data. A cornerstone in ssREAD is the utilization of the Seattle Alzheimer's disease brain cell atlas (SEA-AD) with ID of AD03501, an atlas that includes cells derived from the health and AD human middle temporal gyrus (MTG)[37]. The SEA-AD atlas originally encompassed an extensive dataset of 378,211 cells. We employed the sketch-based analysis feature in Seurat v5 to streamline our analysis, which strategically selects a 'subset' or 'sketch' of 50,000 cells. This analytical decision was driven by achieving computational efficiency while ensuring a robust representation of cellular diversity. Such an atlas painted a detailed cellular tableau, effectively revealing cell distributions of 23 cell types (Fig. 4a). Beyond cell types, the atlas also includes a nuanced statistical breakdown of the cells, distinctly categorized by the Braak stage, Thal phase, gender, and ethnicity, offering a comprehensive glimpse into the atlas's cellular constitution (Fig. 4b–f). As shown, there is no batch

effect among samples regarding Braak stage, Thal phase, and ethnicity. However, there are obvious sex-oriented differences in cell clusters, which may contribute to the possible pathological sex-bias differences in AD.

DEG analysis between AD and control cells revealed distinctive signatures in each cell type (Supplementary Data 8). For example, DEGs associated with the homeostatic state of microglia (e.g., *P2RY12*, *CSF1R*, *CX3XR1*, *TGFBR1*, *MEF2A*, and *ENTPD1*) are decreased, while DEGs associated with the dyshomeostatic state of microglia (e.g., *CTSD*, *APOE*, *AXL*, *SPP1*, and *GPNMB*) are increased in AD compared to the control[30,38] (Fig. 4g). To showcase the feasibility and power of our analysis, we further included a large AD dataset from the prefrontal cortex (PFC) region[39], and integrated the MTG sample with the PFC samples. The top 25 microglia DEGs between AD and control in the integrated data include most genes found in the individual datasets (Supplementary Data 9–11). We also identified 68 upregulated Microglia DEGs between AD and control overlapped across all datasets (Fig. 4h, Supplementary Data 12, and Source Data 4), indicating many DEGs can be recapitulated by using the integrated datasets. Importantly, our comparison highlights the differences in DEGs due to region differences, in which the MTG encompasses many identified DEGs compared to the PFC. This showcases the underlying transcriptomic changes may be more prominent in regions that are affected earlier in Alzheimer's disease in comparison to regions such as the PFC which are affected later in disease. In addition, the integrated data set reveals 17 DEGs that are not present in either individual dataset. The 17 DEGs suggest there are potential transcriptomic changes that are independent of region-based characterization which could only be identified by integrating such datasets. These genes include *DOPEY2*, *CSF2RA*, and *SRRM2* which have been previously linked to AD[40]. For example, *CSF2RA* was upregulated in mouse microglia that have internalized Aβ plaques, and have been treated with IL-1β[40]. *DOPEY* was previously identified in cases (AD and/or MCI) in the ADNI and NIA-LOAD/NCRAD Family studies[41]. Lastly, *SRRM2* was previously studied due to its association with the progression of tauopathy in transgenic mice[42] as well as recruitment to tau aggregates[43]. We compared our DEG results from the integrated sc/snRNA-seq dataset to previously published spatial datasets[9]. Out of the 68 genes that are upregulated in the microglia population, we identified 29 genes (~43%) overlapped with DEGs from previously published spatial transcriptomics datasets (Supplementary Data 13).

Bridging the understanding between single-cell and spatial data, ssREAD was instrumental in performing a cell-type deconvolution analysis between the ST datasets and the SEA-AD atlas using the CARD R package[44]. This synergetic approach underscored the presence and dynamics of critical cell types, including oligodendrocytes, astrocytes, and endothelial cells, across both AD (ST01103) and the control (ST01101) datasets. The cellular insight was further illuminated by visualizing the expression of marker genes, such as *GFAP* (Fig. 4i) for astrocytes (Fig. 4j) and *MOBP* (Fig. 4k) for oligodendrocytes (Fig. 4l), distinguishing between control and AD conditions. Unfortunately, we observed a very low proportion of the microglia cell type in both Visium samples (Supplementary Fig. 3B). We further compared DEGs between AD and control in MAPLE cluster 1 to the identified single-cell DEGs from the integrated AD035 datasets. Among the overlapped DEGs, we highlight two upregulated (*GJA1* and *MT-ATP8*) (Fig. 4m, n) and two downregulated DEGs (*IFITM3* and *TUBB2B*) (Fig. 4q, r) in the astrocyte population and their spatial distribution. Importantly, *GJA1* has been identified as a key regulator in AD pathogenesis, being associated with AD amyloid, tau pathology, and cognitive functions. While depletion of astrocytic *GJA1* is linked to neuroprotection in neurons[45]. In Oligodendrocytes, we highlight four upregulated DEGs, *ERBIN*, *GPRC5B*, *MID1IP1*, and *SLC44A1* (Fig. 4o,p,s,t). Notably, *ERBIN* and *MID1IP1* have been identified in previous datasets as upregulated in Oligodendrocytes for AD pathology cases[46]. The seamless fusion of

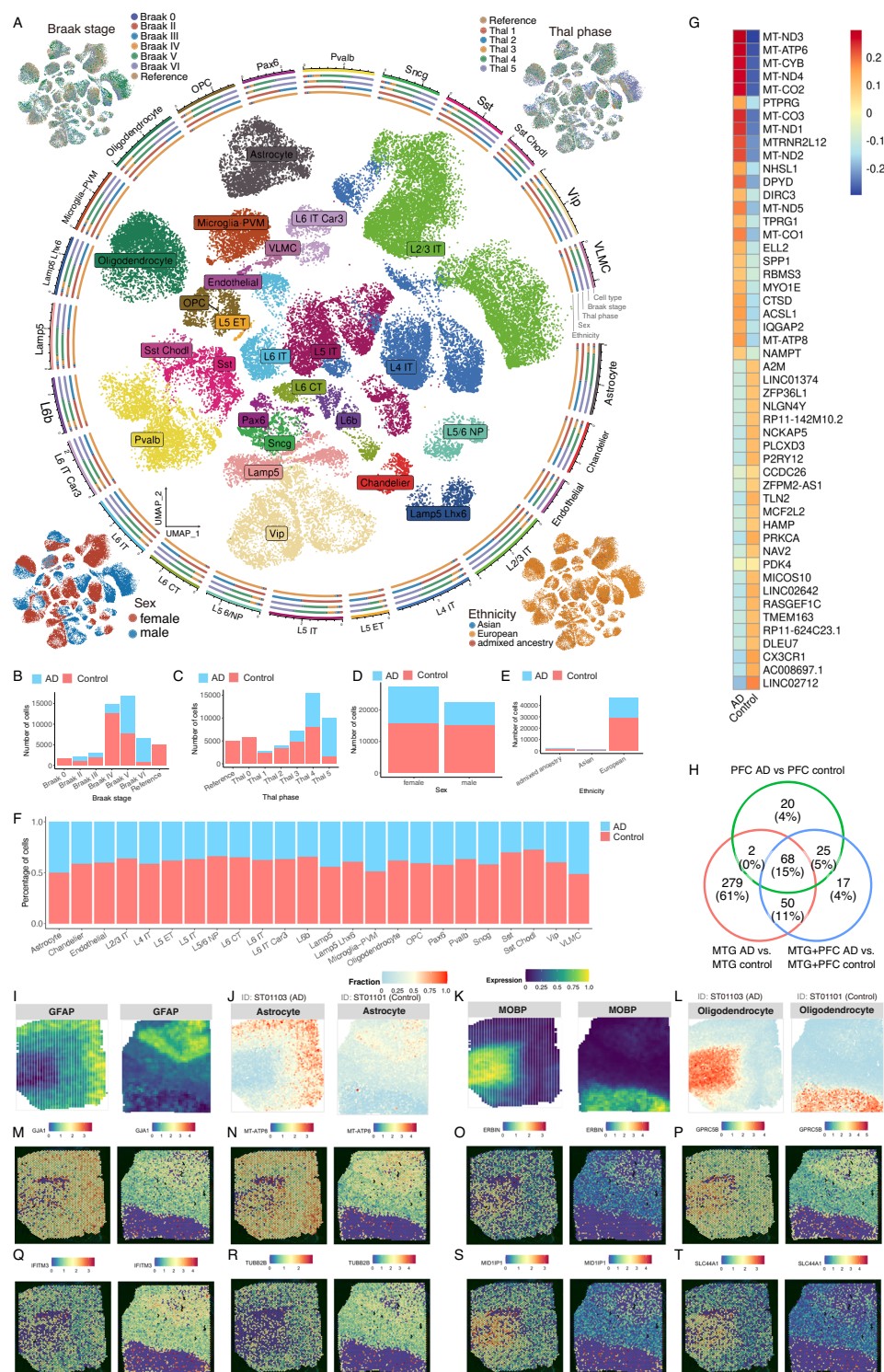

**Fig. 4 | ssREAD facilitates comprehensive spatial transcriptomics analyses synergized with scRNA-seq datasets. a** UMAP representation of cell types derived from the Seattle Alzheimer's Disease Brain Cell Atlas (SEA-AD). Clockwise from the top left, corner insets elucidate the Braak stage, Thal phase, ethnicity, and sex attributes. **b**–**e** Bar plots visualizing the distribution of cells based on the Braak stage, Thal phase, sex, and ethnicity, categorized by the condition in the atlas. **f** Bar plots displaying the fractional representation of cell types, contrasting AD and control within the SEA-AD atlas. **g** Heatmap of top and bottom 25 DEGs identified between AD and control samples in the integrated AD035 MTG dataset in Microglia. **h** Comparison of DEGs overlap in Microglia among AD035 MTG, AD048 PFC, and

their integrated datasets. **i**–**l** Deconvolution analysis of cell types between ST samples (ST01101 and ST01103) and the scRNA-seq SEA-AD atlas, showing the cell type fractions for Astrocytes (**j**) and Oligodendrocytes (**l**). Marker gene expression indicators, *GFAP* for Astrocytes (**i**) and *MOBP* for Oligodendrocytes (**k**). **m**–**t** Spatial distributions of DEGs in AD and control samples within MAPLE cluster 1, cross-validated with single-cell DEGs from the integrated AD035 datasets. Upregulated DEGs in Astrocytes *GJA1* (**m**) and *MT-ATP8* (**n**). Downregulated DEGs in Astrocytes *IFITM3* (**q**) and *TUBB2B* (**r**). Upregulated DEGs in Oligodendrocyte *ERBIN* (**o**), *GPRC5B* (**p**), *MID1IP1* (**s**), and *SLC44A1* (**t**).

snRNA-seq and spatial transcriptomics data, as facilitated by ssREAD, positions this integrated methodology as a potent tool for in-depth cellular analysis, demonstrating the platform's pivotal role in multi-dimensional data integration and offering transformative insights into AD[45]. As a result, we showcase the capability of ssREAD in integrating sc/snRNA-seq and ST data with computational tools like MAPLE, shedding light on the multi-dimensional dynamics of spatially informed subpopulations in AD. By unveiling the biological pathways and regulatory networks associated with disease progression, spread allows a comprehensive understanding of the cellular and molecular landscape in AD, thus bringing us a step closer to unraveling the mysteries of this complex neurodegenerative disease.

## Unveiling Sex-Specific Differences in Alzheimer's Disease at the Cellular Level

Besides the data-driven analysis that can be elaborated from ssREAD, we also showcase the ability to generate biological hypotheses that can be powered by the database. Capitalizing on the wealth of information offered by sc/snRNA-seq data, our investigation delves into the intricate, sex-specific nuances of AD at the cellular level. Using ssREAD, we were able to elucidate the AD heterogeneity between male and female, an aspect not previously pursued in the original research, which primarily focused on the molecular landscape of the over 183k cells in human brain hippocampus vasculature in AD[47] (AD019). Our investigation utilized all 16 samples from the original study (Supplementary Data 14). Commencing with a UMAP visualization of the single-cell data, color-coded by cell type (Fig. 5a), we achieved a broad perspective of 13 cell types (i.e., Arterial cell, Astrocyte, Capillary cell, Ependymal cell, Fibroblast, Microglia, Neuron, Oligodendrocyte progenitor cell, Oligodendrocytes, Pericyte, Smooth muscle cell, T cell, and Venous). From the ensuing breakdown of the proportion and count of each cell type by sex (Fig. 5b–c), we observe more Oligodendrocytes, Astrocytes, and Microglia in overall female cell types than male. Such sex-specific differences in cellular composition raise intriguing questions about the potential roles of these variations in disease pathogenesis and progression, warranting further investigation. We further compared DEGs across four demographic groups: Male AD patients, Female AD patients, Male controls, and Female controls. The result showcased both unique and shared gene signatures among these groups (Fig. 5d). Our findings indicate that both sex and disease status can shape the transcriptomic landscape of cells, which could have profound

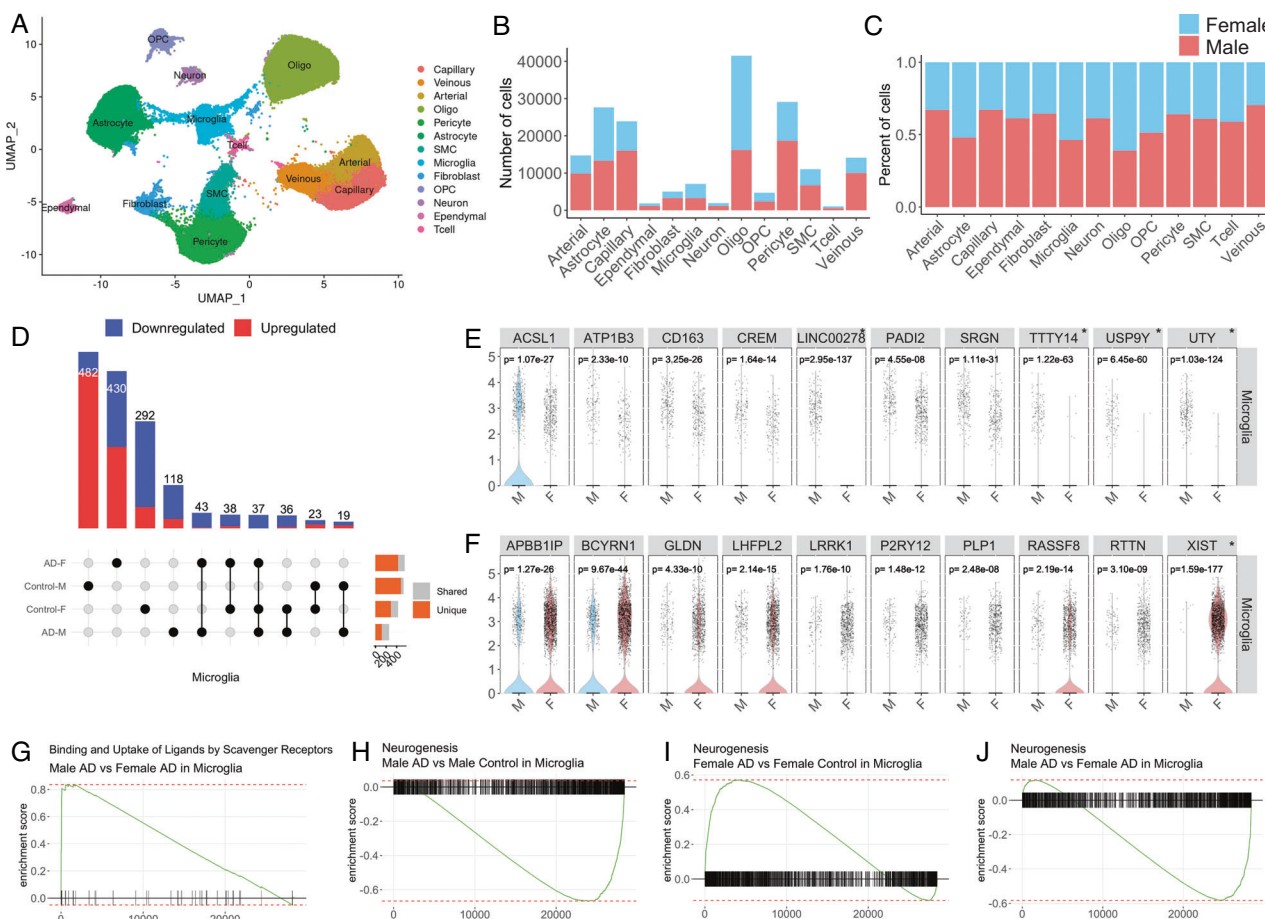

**Fig. 5 | Exploration of sex-specific differences at the cellular level in AD. a** UMAP visualization of the single-cell data used in the analysis, with different cell types color-coded. **b–c** Bar plots illustrate the count and proportion of each cell type, segregated by sex. This reveals any potential differences in cellular composition between male and female samples. **d** UpSet plot showing the unique and shared DEGs across four groups: Male AD patients, Female AD patients, Male controls, and Female controls. **e** Violin plots for the top 10 upregulated DEGs between male and female in microglia. * Indicates sex-chromosomal genes. *p*-values were calculated based on a two-sided Wilcoxon Rank-Sum test and adjusted using Bonferroni correction. **f** Violin plots for the top 10 downregulated DEGs between male and female in microglia. *p*-values were calculated based on a two-sided Wilcoxon Rank-Sum test and adjusted using Bonferroni correction. **g** Gene Set Enrichment Analysis (GSEA) plot showing the enrichment of genes involved in binding and uptake of ligands by scavenger receptors. **h–j** GSEA plots showing the enrichment of genes involved in neurogenesis for three different comparisons: Male AD vs. Male Control in Microglia (**h**), Female AD vs. Female Control in Microglia. (**i**), and Male AD vs. Female AD in Microglia (**j**). These plots highlight the sex-specific differences in neurogenesis-related gene activity under AD conditions.

implications for understanding the underlying molecular mechanisms of AD and developing targeted therapies.

By comparing the gene expressions between males and females, we revealed unique sex-specific gene signatures within different cell types (Supplementary Data 15). Specifically, we identified 44 upregulated and 108 downregulated genes in male microglia vs female (Fig. 5e). We highlight the top 10 upregulated and downregulated DEGs, in which we identify *ATP1B3*, a previously identified ARM and DAM gene is downregulated in our comparison of female microglia to male microglia[29,30]. On the other hand, *LHFPL2*, and *RTTN* are DAM genes that we find are upregulated in female microglia compared to male[30]. Our data also remains consistent with a previous publication of AD-related transcriptional sex differences that identified *ACSL1*, *ATP1B3*, *CD163*, *CREM*, and *SRGN* as downregulated in female microglia cell nuclei compared to male[48]. The upregulation of genes *APBB1IP*, *GLDN*, *LHFPL2*, *LRRK1*, *P2RY12*, *RASSF8*, and *RTTN* in female microglia compared to male is also consistent with previous findings[48]. We validated the sex related DEGs findings using two additional datasets (ID AD035 and AD048) (Supplementary Data 16-17 and Supplementary Fig. 5). We find that the Y chromosomal genes *UTY*, *LINC00278*, *TTTY14*, and *USP9Y* are found in both datasets and correspond to genes that are upregulated in male microglia; While the X chromosomal gene, *XIST*, is upregulated in female microglia and is found in both datasets. Our analysis of Gene Set Enrichment Analysis (GSEA) further explored the activity of binding and uptake of ligands by scavenger receptors, which are known to be involved in the recognition and clearance of various ligands, including modified lipoproteins, cellular debris, and pathogens[49]. In AD, the dysregulation of scavenger receptor activity has been implicated in the clearance of Aβ plaques (Fig. 5G), which are one of the hallmark pathological features of the disease[50]. Previous studies have reported sex differences in scavenger receptor expression in the context of AD[51]. Additionally, we observed differences in GSEA enrichment of neurogenesis pathways in Microglia under AD conditions. Microglia can shape adult hippocampal neurogenesis[52]. We compared Male AD vs. Male Control (Fig. 5H), Female AD vs. Female Control (Fig. 5I), and Male AD vs. Female AD (Fig. 5J). Interestingly, neurogenesis pathway activity varied significantly, with upregulation observed in female AD, and downregulation in male AD patients. This observation denotes a sex-dependent dysregulation of neurogenesis in AD, indicating that the disease could affect foundational neural processes in a sex-specific manner. Enhancing our understanding of how microglial activation states differentially regulate adult neurogenesis in men and women could yield invaluable insights into the disease's intricate mechanisms. Taken together, these results underscore the intricate interplay between sex and cellular and molecular profiles in AD. They highlight the necessity of considering sex as an integral factor in AD research, and point towards the potential for developing more personalized, sex-specific therapeutic strategies in the future.

## Discussion

We develop ssREAD, a single-cell and spatial RNA-seq database for AD, that not only accommodates the expansion of sc/snRNA-seq data but is uniquely poised to incorporate and harness the emergent wealth of AD-related ST data. ssREAD houses data encompassing various species, diseases, tissues, and cell types, thereby permitting granular analyses that could reveal intricate biological phenomena. ssREAD's capabilities are broad, allowing for diverse analytical activities such as contrasting diseased tissues against healthy controls, identifying distinct cell types through gene markers, and examining gene expression across diverse tissues and cell types. The most compelling findings of sex differences in AD unveil a profound heterogeneity between male and female cellular profiles. Particularly, females exhibited a heightened presence of Oligodendrocytes, Astrocytes, and Microglia

compared to males. Furthermore, while female AD patients showed upregulation in neurogenesis pathway activity, their male counterparts displayed a stark downregulation. These insights align with the findings of previous studies claiming that transcriptional responses were substantially different between sexes in different cell types[46], underscoring the pivotal role of gender in shaping AD's cellular and molecular features. Last but not least, designed with the end-user in mind, ssREAD is characterized by its user-friendly interface, query function, and visualizations. In terms of its infrastructure, ssREAD utilizes high-performance computing to manage large-scale single-cell data analysis efficiently. It comprises a cutting-edge server architecture employing diverse programming languages, machine learning frameworks, and data visualization libraries. As our understanding of the complexity and heterogeneity of biological systems continues to deepen, tools like ssREAD can play an increasingly vital role in our pursuit and understanding of AD studies.

To ensure the stability and availability of ssREAD, we have set up backup servers and implemented cloud-based backup solutions. In case of unavailability of the main server, our development team can switch the link to redirect to an alternative server without requiring users to enter a different URL. To improve transparency and provide real-time updates for our users, we have created a status page for ssREAD (https://ssread.statuspage.io/). This page features automated monitoring of the server's status and allows users to track the operational status of the database, including any scheduled maintenance or unexpected downtime. To widen the accessibility and integration of ssREAD, we intend to develop an R package within the Bioconductor project and a Python library in the future, enabling users to access all datasets both locally and remotely through ssREAD's server-side API. We plan to update the ssREAD database regularly, with new data and features being added every six months. This will ensure that our users have access to the most recent and relevant data in the field of single-cell research. Recognizing the need for a platform that accommodates the diverse range of interests in the scientific community, we plan to include a wider range of sc/snRNA-seq and spatial omics data of neural systems such as human iPSC-derived neurons, glia, and neural organoids. Furthermore, we aspire to include more neurodegenerative diseases like Frontotemporal lobar degeneration, Parkinson's disease, and Amyotrophic lateral sclerosis, serving a broader research community. Future plans also include expanding our analytical pipelines and visualization methodologies to enrich the capabilities of ssREAD.

## Methods

### ScRNA-seq and snRNA-seq data categorization

In constructing our comprehensive atlas, we meticulously curated a collection of 67 studies that encompass a diverse array of brain regions, spanning both sexes and a wide age range in human and murine models. This extensive compilation yielded a dataset that encapsulates a rich tapestry of cellular profiles. We have methodically reclassified the original datasets into distinct subsets, each meticulously delineated by species (human or mouse), gender (male or female), brain region (including but not limited to the cortex, Middle temporal gyrus, superior frontal gyrus, cerebellum, subventricular zone, superior parietal lobe, and hippocampus), pathological status (disease or control), and age bracket (7, 15, or 20 months for mice; 50–100+ years for humans) (Supplementary Data 18).

### ST data preprocessing

Spatial transcriptomics data preprocessing was performed using the Seurat v5 and Squidpy v1.3.0 packages. Seurat was employed for quality control, normalization, and identification of highly variable genes[53]. Squidpy was used for spatially-resolved computations, such as spatial autocorrelation analysis, extracting neighborhood information, and spatially-resolved clustering[54].

## Spatially variable genes (SVGs) identification

To identify spatially variable genes, the spaGCN v1.2.7 was utilized[13]. It integrates graph convolutional network (GCN) with spatial transcriptomics to detect genes that vary significantly across different spatial locations in tissue samples.

## Cell type annotation

We applied a similar cell-type annotation strategy used in scREAD. Our quest for analytical precision led us to modify our initial cell-type annotation strategy to increase annotation accuracy. We recognized that several marker genes used in the original release demonstrated sub-optimal specificity. To address this, we executed two iterations of cell type annotation, which involved filtering out several less specific markers. During the first iteration, only Neurons were annotated. Following this, the Neurons were then isolated from the complete dataset, and second-iteration marker genes were employed to annotate Excitatory neurons and Inhibitory neurons. The cell labels for Neurons were subsequently replaced by those of Excitatory neurons or Inhibitory neurons. This revised workflow enhances the annotation quality by accounting for the discrepancies between different cell types and subtypes. The marker gene list can be accessed in Supplementary Data 19.

We used SCINA R package that leverages prior marker gene information and simultaneously performs cell type clustering and assignment for known cell types[55]. Furthermore, SCINA shows good performances among prior-knowledge classifiers when high-quality marker genes are provided[56]. Each cell was assigned a cell type based on a manually created marker gene list file using SCINA v1.2.0, whereas the cells with unknown labels marked by SCINA were first compared with predicted clusters from Seurat, and then the unknown labels were assigned to the most dominant cell types within the predicted clusters.

## Cell type deconvolution analysis

Deconvolution of mixed cell populations was performed using the CARD v1.1 R package[44], which employs a reference-based approach to estimate the proportions of different cell types in bulk RNA-seq or spatial transcriptomics data. It uses cell-type-specific gene expression signatures from a reference dataset to calculate cell-type proportions.

## Differentially expressed gene and functional enrichment analysis

DEG analysis was performed using the MAST v1.28.0 R package[57] through Seurat's FindMarkers function. This function detects DEGs between various conditions, such as disease versus control or male versus female. For overall AD versus control comparisons, age at death and sex were included as covariates. For sex-specific differences in AD versus control comparisons, age at death was included as the covariate. The p-values obtained from each comparison were adjusted to control the false discovery rate (FDR) using the Bonferroni correction for all genes in the dataset. Genes with an FDR < 0.05 were considered differentially expressed. The enrichment analysis was conducted using Enrichr's API services to identify relevant Gene Ontology (GO) terms and Reactome pathway databases[58,59]. The statistical significance of the enrichment was determined using a hypergeometric test which is a binomial proportion test that assumes a binomial distribution and independence for the probability of any gene belonging to any set. The p-values obtained from Enrichr were adjusted to q-values using the Benjamini-Hochberg correction method.

## Cell-cell communication analysis

Cellular communication was investigated using the CellChat v2[60] R package toolkit for the inference, visualization, and analysis of cell-cell communication patterns from single-cell and spatial transcriptomics data. CellChat v2 extends to analyze communication patterns among neighboring cell clusters within spatially resolved transcriptomic landscapes, with an expanded communication network interactions database encompassing a comprehensive compendium of over 1,000 interactions.

## Tissue architecture identification

The tissue architecture was analyzed using RESEPT v1[15], a computational tool that generates an atlas of regional gene expression patterns in spatial transcriptomics data. This tool enables the visualization of how gene expression varies across different spatial regions within a tissue sample.

## Gene regulatory network analysis

Regulatory analysis was conducted using DeepMAPS v1.0[33,61]. We employed DeepMAPS to build the cell and gene embeddings and obtained active gene modules through the graph transformer model. The gene modules were further sent to DeepMAPS to perform cell cluster active gene module determination, de novo motif finding, and TF matching and CTSR determination with the default parameters. Gene regulatory networks were constructed to indicate the predicted TF-gene regulatory relations via Cytoscape[62,63].

## User interface

We made substantial improvements to the user interface to enhance accessibility and ease of use. Key modifications were made within the dataset details page, such as the addition of gene expression display for single cells/spots using feature plots and violin plots. We also introduced a dedicated query page for DEG searching. Notably, search results can now be refined based on sex, group, and condition parameters. The new design offers clearer demarcation between DEG searching and query options. Moreover, we integrated the function to calculate overlapping DEGs from multiple comparisons, an approach originally outlined in the scREAD protocol[64]. This functionality is now delivered through an interactive online query. We also redesigned the homepage for better usability, positioning the search bar at the top and displaying key information about the number of species, studies, assays, and versions for higher visibility.

## ssREAD server construction

ssREAD is hosted on an HPE XL675d RHEL system outfitted with a 2 × 128-core AMD EPYC 7H12 CPU, 64GB RAM, and 8×NVIDIA A100 80GB GPU. Our backend server, written in TypeScript and built with the koa.js framework, leverages Auth0 to provide independent user authentication and authorization services. Our frontend is constructed with NUXT, utilizing Vuetify as the UI library and Plotly.js for data visualization. Communication between frontend and backend servers is enabled using REST API. This streamlined server construction ensures robust, efficient, and scalable performance of ssREAD.

## Reporting summary

Further information on research design is available in the Nature Portfolio Reporting Summary linked to this article.

# Data availability

All data used were sourced from public collections, and are detailed with online accession numbers in Supplementary Data 1. ssREAD is freely available at https://bmblx.bmi.osumc.edu/ssread/. A backup link is also provided at https://go.osu.edu/ssread. Relevant raw data from each figure is available in the Source Data file. The processed data in this study can be downloaded through the link https://bmblx.bmi.osumc.edu/ssread/downloads. Source data are provided in this paper.

# Code availability

The frontend code is available at https://github.com/OSU-BMBL/ssread. The backend code is available at https://github.com/OSU-BMBL/ssread-backend. Additionally, a real-time ssREAD server status

page is available at https://ssread.statuspage.io/. The source code to the version of the code used in this study is also available on Zenodo[65].

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

## Acknowledgements

This work was supported by awards R21HG012482 (Q.M.) and R01AG075092 (H.F.) from the National Institutes of Health (NIH). We would like to acknowledge Qi Guo, Shaohong Feng for their kind help in testing the website and database. We also thank Mr. Nicholas Sweeney and Ms. Tae Yeon Kim for their helpful discussions during the revision.

## Author contributions

Q.M. and H.F. designed the manuscript contents and experiments. C.W. and M.M. contributed to data collection and analysis. C.W. developed and implemented the database and the API. D.A., M.M., and A.M. tested the database and interpreted the results. C.W., D.A., A.M., and J.B. drafted the manuscript. All authors revised the final manuscript.

## Competing interests

The authors declare no competing interests.
