## [Peer Review File · Nature Communications]

A Single-cell and Spatial RNA-seq Database for Alzheimer's Disease (ssREAD)REVIEWER COMMENTS

Reviewer #1 (Remarks to the Author):

In the manuscript by Wang et al., the authors developed an upgraded platform, called ssREAD, that integrates single cell/single nucleus RNA-seq and spatial transcriptomics datasets of Alzheimer's disease from both mouse models and human patients. This platform includes processing and analyzing of raw data, as well as a user-friendly interface for the community to explore. Although ssREAD is a convenient resource for AD researchers to mine datasets, there lacks a functional application of the platform. The manuscript is merely an introduction of the method. Without further usage of the resource as an example, the manuscript is insufficient for publication.

1. The manuscript would be hugely improved if the authors can draw some conclusions from the integrated data. For example, what are the top DEGs between AD and control in the integrated data? How do they compare with previously published datasets? What is the spatial distribution of these DEGs? How do DEGs identified from sc/snRNA-seq correlate with those found from spatial transcriptomics?

2. It is surprising to see that in Figure 3G, immune related pathways are downregulated in AD, compared to control, since microglia have been shown to be activated in AD. Please discuss about the results.

3. Both ATF6 and EGR1 are stress response-related genes. Please check whether the genes in the regulons of ATF6 and EGR1 are stress-related genes. Please clarify in which cell types are the TFs (such as NR1H3, ATF6, EGR1, HIF1A) identified from spatial transcriptomics expressed.

4. For Figure 5E, please provide a rationale why these 10 genes are selected and explain the color code. For example, LYVE1 is a marker for lymphatic endothelial cells and border-associated macrophages. Why is LYVE1 included in the figure? Also, please provide p values for the violin plots.

Reviewer #2 (Remarks to the Author):

Wang et al. developed ssREAD, a Single-cell and Spatial RNA-seq Database for Alzheimer's Disease, which is an update version of scREAD. ssREAD offering both breadth and depth of sc/snRNA-seq and spatial transcriptomics data that stand unrivaled compared to other available resources or databases. However, there are many issues should be addressed before its publication.

Comments about the manuscript:

1. As I know, TACA (<https://alz-journals.onlinelibrary.wiley.com/doi/full/10.1002/trc2.12350>) is also a scRNA-seq database for AD, it should be mentioned and compared in your manuscript. In addition, these similar databases were cited in figure 1, in my opinion, they should be firstly introduced in the introduction part to help readers get the research status of this filed properly.

2. I noticed that ssREAD includes more scRNA-seq studies than SCAD-Brain, but with less samples. Did you filtered some samples by quality control or other purpose?

3. Table S2 is cited in line 129, but it is the marker genes used to as a cell to a specific cell type, the citation seems be wrong.

4. The results interpretation is too brief and macroscopic. For example, Figure 3A-3D provides rich information but the lines 167 to 183 do not mention the details, just came to a conclusions like "provide a precise delineation of the tissue architecture and functional zones in both control and AD brain tissues". In the reviewer's opinion, this makes it difficult for readers to accurately interpret the results in the article and web.

5. The authors selected two ST data (ST01101 and ST01103) to give us a case study to show cellular heterogeneity in AD. From the ssREAD, I noticed that ST001 study has 12 samples, why did you select only these two samples for a case study? Can these DEGs or SVGs or regulons be consistently found in other samples?

6. Which datasets were used to determine the Sex-Specific Differences in Alzheimer's Disease at the Cellular Level? Please provide the ID in the manuscript. Additionally, the distribution of age in selected samples should be provided to avoid age confounding the results. Similarly, can these sex-specific genes be validated by other datasets?

7. Developing a free database, reducing barriers and speeding up single-cell research are the great undertaking. The maintenance of a web server is a difficult task. Therefore, the publication of a paper developed a tool is not the end, but the beginning. However, I noticed that scREAD lost maintenance for a long time after it was released. Have you taken any measures to ensure the maintenance of ssREAD this time?

Issues for the web server.

1. The main functions and analyses that ssREAD performed are comparisons and pathway enrichment. But there are many types of analyses designed for scRNA-seq and ST data, cell communication for example. Further in-depth analysis could be included in ssREAD.

2. It is nice that the ssREAD provides users variable arguments. But when users change an argument, there is no loading effect to remind users. The reviewer suggest that it should be added.

3. For download of the results figures, the text distribution, resolution, and size of the snapshot are not that high quality. This would make it difficult for users to put it directly in their paper.

4. Additionally, necessary explanation should be added to improve users' sense of experience. For example, if it is not suitable for a sample to perform Disease vs disease (same region) or Disease vs. disease (different region), you should remind the users and give it a reason.

5. DEGS in disease and control is performed between the selected sample and another sample.

Whether you performed sample integration to provide real group comparisons? Biological repeatability will improve the reliability of expression difference analysis. Comparing only two samples will lose the advantage of sample size in the original study.

6. Detailed methods should be provided in the web server. As a web server, the methods description is important for users to replicate the results and use the results in their publications.

7. After clicking the EXAMPLE button, the functional part under the QUERY module "Find overlapping DEGs from multiple comparisons" is loading all the time and no results are returned.

8. When searching for differentially expressed genes, the gene name search field should allow the partial match as well, for example, if the user only type 'Apo', it should match with 'ApoE' gene name.

9. The speed of the ssREAD is slow. It is better to speed up it to make it more user friendly.

Reviewer #3 (Remarks to the Author):

This is a meaningful work, and research on Alzheimer's Disease indeed requires continuously updated databases to support it, especially with the development of spatial transcriptomics and single-cell technologies, as well as their applications in neuroscience.

After reading the article:

1. The article mentions the comparison between ssREAD and other similar databases, and I checked the table including Single-cell portal and STOMicsDB, and found that the numbers may not be accurate, which could be due to the information being updated until May 2023, and it may need to be confirmed again;

2. I compared ssREAD with scREAD and found that ssREAD has made many improvements in UI/UX, but the article also mentions some analyses done after incorporating spatial transcriptomics data. It would be helpful to provide some online analysis tools on the website, as currently it mainly focuses on static page display and visualization.

Reviewer #1

In the manuscript by Wang et al., the authors developed an upgraded platform, called ssREAD, that integrates single cell/single nucleus RNA-seq and spatial transcriptomics datasets of Alzheimer's disease from both mouse models and human patients. This platform includes processing and analyzing of raw data, as well as a user-friendly interface for the community to explore. Although ssREAD is a convenient resource for AD researchers to mine datasets, there lacks a functional application of the platform. The manuscript is merely an introduction of the method. Without further usage of the resource as an example, the manuscript is insufficient for publication.

Response: Thank you for your consideration of our ssREAD platform as a convenient resource for AD researchers. Indeed, our goal is to facilitate the use of present and future datasets available to the AD community. We hope to encourage the generation of new guided hypotheses that may ultimately become new therapeutic targets. To highlight the functional application of the platform, we have selected to integrate two of the largest available sc/snRNAseq datasets. We have further explored the outcomes of integration by comparing the integrated data to previously published datasets. Importantly, we show that integrated datasets can reveal important similarities between datasets, which facilitates the formation of stronger hypotheses. To address your helpful comments, we have focused on the upregulation of genes within the microglia population and compared region-difference and Braak stage-difference besides the original sex difference. We believe adding this data will not only appropriately address your comments and questions, but also strengthen the focus of our manuscript.

1. The manuscript would be hugely improved if the authors can draw some conclusions from the integrated data. For example, what are the top DEGs between AD and control in the integrated data? How do they compare with previously published datasets? What is the spatial distribution of these DEGs? How do DEGs identified from sc/snRNA-seq correlate with those found from spatial transcriptomics?

Response: Thank you for your constructive comments. To showcase the feasibility and power of our updated platform, we have integrated two of the largest available sc/snRNAseq datasets (SEA-AD [1] and Mathys *et al.*, Cell 2023 [2]). It is important to note that these two datasets consist of two different brain region samples, i.e. the MTG and the prefrontal cortex (PFC). We performed DEG analysis for the microglia cell-type population using the integrated dataset and previously published datasets (MTG: **Supplementary Table S8**; PFC: **Supplementary Table S9**; integrated: **Supplementary Table S10**). The top 25 DEGs in the integrated dataset include most genes found in the individual datasets (**Figure 4G** and **Supplementary Table S11**). We also compared all the upregulated Microglia DEGs between AD and control in each dataset to each other (**Figure 4H** and **Supplementary Table S12**). We identified 68 genes overlapped across all datasets, indicating many DEGs can be recapitulated by using the integrated datasets. Importantly, our comparison highlights the differences in DEGs due to region, in which the MTG encompasses many identified DEGs compared to the PFC, showcasing underlying transcriptomic changes may be more prominent in regions that are affected earlier in Alzheimer's disease in comparison to regions such as the PFC which are affected later in the disease. In addition, the integrated data set reveals 17 new DEGs that are not present in either individual dataset. The 17 DEGs suggest there are potential transcriptomic changes that are independent of region-based characterization, which could only be identified by integrating such datasets.

Furthermore, we compared our DEG results from the integrated sc/snRNA-seq dataset to previously published spatial datasets [3]. Out of the 68 genes upregulated in the microglia population, we identified 29 genes (~43%) overlapped with DEGs from previously published spatial transcriptomics datasets (**Supplementary Table S13**). Unfortunately, our cell type deconvolution analysis using SEA-AD snRNA-seq data and Visium spatial transcriptomics datasets from human MTG reveals a very low proportion of microglia cell type in our Visium samples (**Supplementary Figure S3B**). When we looked at the spatial distribution of each gene of 29 upregulated microglial genes or a gene module of those 29 genes, we did not find distinct patterns between AD and Control samples, probably due to the lack of microglia-enriched spots in our Visium samples. Interestingly, we did find distinct spatial distributions of several DEGs in astrocytes and oligodendrocytes (**Figures 4M-T**). This data

suggests that astrocytes and oligodendrocytes may also contribute DEGs and enriched pathways from MAPLE Cluster 1 besides microglia, although they have to be validated by single-cell spatial transcriptomics such as 10x Genomics Xenium, Vizgen MERSCOPE, NanoString CosMx.

[1] Gabitto M, *et al.* Integrated multimodal cell atlas of Alzheimer's disease. *Res Sq*, (2023).

[2] Mathys H, *et al.* Single-cell atlas reveals correlates of high cognitive function, dementia, and resilience to Alzheimer's disease pathology. *Cell* 186, 4365-4385.e4327 (2023).

[3] Chen S, *et al.* Spatially resolved transcriptomics reveals genes associated with the vulnerability of middle temporal gyrus in Alzheimer's disease. *Acta Neuropathologica Communications* 10, 188 (2022).

2. It is surprising to see that in Figure 3G, immune related pathways are downregulated in AD, compared to control, since microglia have been shown to be activated in AD. Please discuss about the results.

Response: Thank you for your comments. We have further investigated the association of the genes within each pathway to microglial states previously published by Sun *et al.*, as well as disease-associated microglia (DAM) and activated response microglia (ARM) genes [1-3]. In agreement with this previously published data, three out of five of our upregulated pathways (Regulation of Expression of Slits and Robos, Selenoamino Acid Metabolism, Eukaryotic Translation Elongation) include several genes that make up a population of microglia (MG3) that is highly enriched with disease-associated microglial genes. On the other hand, the five downregulated pathways (Scavenging of Heme from Plasma, Interleukin 10 Signaling, Tnfs Bind their Physiological Receptors, Fcgr Activation, Creation of C4 and C2 Activators) are composed of genes found in microglia states that are associated with inflammation due to presence of cytokine and cytokine receptor-related genes (**Supplementary Figure S4**). Therefore, the upregulated pathways overlap with disease-associated microglia states, while downregulated pathways do not include genes found in disease-associated microglia states. In line with previous findings, inflammatory states of microglia change in relation to AD progression, with more strongly correlated inflammatory states present at early disease stages and weaker inflammatory states present at late disease stages. Therefore, the downregulation of inflammatory-related states in our data coincides with the dysregulation of microglia states found in AD. Importantly, the dataset used for our analysis is composed of three control and three human AD samples, which yielded very few downregulated DEGs. Therefore, integrating this dataset with other coming AD spatial transcriptomics datasets may be beneficial and further highlights the importance of the current ssREAD database and our manuscript.

[1] Sun N, *et al.* Human microglial state dynamics in Alzheimer's disease progression. *Cell* 186, 4386-4403.e4329 (2023).

[2] Keren-Shaul H, *et al.* A Unique Microglia Type Associated with Restricting Development of Alzheimer's Disease. *Cell* 169, 1276-1290.e1217 (2017).

[3] Sala Frigerio C, *et al.* The Major Risk Factors for Alzheimer's Disease: Age, Sex, and Genes Modulate the Microglia Response to A β Plaques. *Cell Rep* 27, 1293-1306.e1296 (2019).

3. Both ATF6 and EGR1 are stress response-related genes. Please check whether the genes in the regulons of ATF6 and EGR1 are stress-related genes. Please clarify in which cell types are the TFs (such as NR1H3, ATF6, EGR1, HIF1A) identified from spatial transcriptomics expressed.

Response: Thanks for your suggestion. We performed enrichment analysis using GO and REACTOME databases to identify the pathways of ATF6 and EGR1 regulons. Our results showed that genes, including XBP1, HSPA5, DDIT3, SEL1L, ATP2A2, and HSP90B1, regulated by ATF6 are related to response to ER stress pathways (**Supplementary Table S7**). While, 106 genes regulated by EGR1 are related to stress response, such as CDKN1A, MYC, TP53, and RXRA, which are related to pathways including oxidative stress, ER stress, chemical stress, and stress-activated MAPK cascade. All TFs, including ATF6 and EGR1, were identified in MAPLE cluster 1, which is primarily annotated to Layer 5 and Layer 6 (**Supplementary Figure S3A**). Deconvolution analysis revealed that the major cell type proportions within MAPLE cluster 1 are Astrocytes,

Oligodendrocytes, and Endothelial cells (**Supplementary Figure S3B**). Therefore, the data suggests cell types, other than microglia, may be responsible for stress response-related genes, although we cannot exclude the possibility of microglial contribution due to the lack of microglial enriched spots in our samples. Testing these TFs using more Visium samples and validating them by single-cell spatial transcriptomics such as 10x Genomics Xenium, Vizgen MERSCOPE, and NanoString CosMx are warranted in the future.

4. For Figure 5E, please provide a rationale why these 10 genes are selected and explain the color code. For example, LYVE1 is a marker for lymphatic endothelial cells and border-associated macrophages. Why is LYVE1 included in the figure? Also, please provide p values for the violin plots.

Response: Previously, we selected the top 20 DEGs in microglia based on the Bonferroni-adjusted p -value from low to high, and removed genes in the sex chromosome, leading to 10 genes including LYVE1. We have corrected our selection criteria, to better address our analysis and results. In the revision, we select the top 10 upregulated and downregulated microglia DEGs between male and female based on the rank of logFC values from high to low, respectively. We indicated the sex-related genes with a “*” mark in **Figure 5E**. The color code has been removed to avoid misunderstanding, and Bonferroni-adjusted p -values have been indicated in **Figure 5E** now. All DEGs between male and female in each cell type can be found in **Supplementary Table S15**.

Reviewer #2

Wang et al. developed ssREAD, a Single-cell and Spatial RNA-seq Database for Alzheimer's Disease, which is an update version of scREAD. ssREAD offering both breadth and depth of sc/snRNA-seq and spatial transcriptomics data that stand unrivaled compared to other available resources or databases. However, there are many issues should be addressed before its publication.

Comments about the manuscript:

1. As I know, TACA (<https://alz-journals.onlinelibrary.wiley.com/doi/full/10.1002/trc2.12350>) is also a scRNA-seq database for AD, it should be mentioned and compared in your manuscript. In addition, these similar databases were cited in figure 1, in my opinion, they should be firstly introduced in the introduction part to help readers get the research status of this filed properly.

Response: Thank you for your comments. We have updated the numbers and revised our manuscript for the TACA database and other data resources in **Figure 1F**. We also added a paragraph in the Introduction Section to introduce and compare the features of ssREAD with these databases.

2. I noticed that ssREAD includes more scRNA-seq studies than SCAD-Brain, but with less samples. Did you filtered some samples by quality control or other purpose?

Response: Thank you for your comments. No, we did not filter out samples for any purpose. The term 'dataset' in ssREAD encompasses multiple samples in the same study with the same conditions. For example, AD035 is the ID of the study [1], which involves 84 samples. We grouped the 84 samples into 12 datasets based on sample conditions. The AD03501 dataset includes 2 samples that are both female AD in Braak stage II, the AD03502 dataset includes 3 samples that are all female AD in Braak stage IV, and so on. Consequently, the number of datasets displayed in ssREAD is fewer than the total number of samples in other databases. We have updated ssREAD website to include a detailed explanation of our dataset processing methodology (<https://bmbx.bmi.osumc.edu/ssread/help/methods>). This includes information on the quality control parameters used, data normalization procedures, and any other relevant processing steps that contribute to the robustness of the datasets included in ssREAD.

[1] Gabitto MI, *et al.* Integrated multimodal cell atlas of Alzheimer's disease. *Res Sq*, (2023).

3. Table S2 is cited in line 129, but it is the marker genes used to as a cell to a specific cell type, the citation seems be wrong.

Response: Thank you for your comments. We have moved reference to this marker gene table to the “cell type annotation” section in Method (Line 548). The table is updated as **Supplementary Table S19** in the revision.

4. The results interpretation is too brief and macroscopic. For example, Figure 3A-3D provides rich information but the lines 167 to 183 do not mention the details, just came to a conclusions like “provide a precise delineation of the tissue architecture and functional zones in both control and AD brain tissues”. In the reviewer’s opinion, this makes it difficult for readers to accurately interpret the results in the article and web.

Response: Thank you for your comments. We have included further interpretation of these results in lines 238-256 of the text. We describe that the resulting spatial delineation in Figure 3C remains comparable to the six cortical layers identified in the original study. However, Layers 5-6 of the AD sample (ST01103) exhibit slight differences in their delineation compared to controls and their original labels, suggesting there may be underlying differences in the functional zones of these layers. The data is consistent with previous publications suggesting layer 5 is highly relevant to changes in AD, including the accumulation of neurofibrillary tau tangles [1-3]. For Figure 3C we also dive into the significance of the differences observed by our MAPLE analysis and their corresponding layer annotations shown in Figure 3D. We show cluster 1 corresponds to layers 5 and 6 of the AD sample (ST01103), but only corresponds to layer 6 of the control sample (ST01101), suggesting deviations in layer 5 of AD cases compared to controls may also be a result of differences in their spatial organization as well as their gene expression patterns.

[1] Arnold SE, Hyman BT, Flory J, Damasio AR, Van Hoesen GW. The topographical and neuroanatomical distribution of neurofibrillary tangles and neuritic plaques in the cerebral cortex of patients with Alzheimer's disease. *Cereb Cortex* 1, 103-116 (1991).

[2] Morrison JH, Hof PR. Selective vulnerability of corticocortical and hippocampal circuits in aging and Alzheimer's disease. *Prog Brain Res* 136, 467-486 (2002).

[3] Braak H, Braak E. On areas of transition between entorhinal allocortex and temporal isocortex in the human brain. Normal morphology and lamina-specific pathology in Alzheimer's disease. *Acta Neuropathol* 68, 325-332 (1985).

5. The authors selected two ST data (ST01101 and ST01103) to give us a case study to show cellular heterogeneity in AD. From the ssREAD, I noticed that ST001 study has 12 samples, why did you select only these two samples for a case study? Can these DEGs or SVGs or regulons be consistently found in other samples?

Response: First, the two ST data (ST01101 and ST01103) are from the ST011 study, which has 6 samples, rather than the ST001 study with 12 control samples (without AD) from the PFC region. In the ST011 study, there are three control and three AD samples from the MTG region, respectively. We randomly selected ST01101 (control sample) and ST01103 (AD sample) for the case study with no specific reason. All the DEG, SVG, and regulon analyses are performed on MAPLE clusters. Due to the fact that the determination of MAPLE clusters highly depends on coupled samples, MAPLE clusters identified in other samples might be incomparable to those identified in ST01101 and ST01103. Thus, it is not feasible to evaluate the consistency of DEGs, SVGs, and regulons at the MAPLE cluster level. Instead, we now also perform DEG analysis between AD (ST01103) and control (ST01101) for each functional layer (**Supplementary Table S2**) and evaluate the DEG consistency with the other two comparisons (i.e., ST01102 vs. ST01104 and ST01106 vs. ST01105) (**Supplementary Tables S3-4**). Results showed that DEGs between AD and control in each layer overlapped among the three comparison groups (**Supplementary Figure S2**).

6. Which datasets were used to determine the Sex-Specific Differences in Alzheimer's Disease at the Cellular Level? Please provide the ID in the manuscript. Additionally, the distribution of age in selected samples should be provided to avoid age confounding the results. Similarly, can these sex-specific genes be validated by other datasets?

Response: Thank you for your comments. We used all the 16 samples originating in [1] with ssREAD ID AD019. We added **Supplementary Table S14** to specify the age, gender, and condition information. Noted that, to mitigate the potential confounding effect of age in our analysis, we have used the MAST R package [2] which can regress out age as a covariate while computing DEGs. This methodology is detailed in the Differential Gene Expression Method section, demonstrating how we minimized the confounding effects of age to ensure the accuracy of our findings regarding sex-specific differences. We now compared our sex-specific DEGs identified in the AD019 (hippocampus) with the other two studies (MTG: AD035 and PFC: AD048) (**Supplementary Tables S16-17**). The cross-validation results are not ideal (**Response Figures A1-G2 listed below**) thus not included in the revised manuscript. We reasoned such differences are induced by the heterogeneity of different brain regions.

[1] Yang AC, *et al.* A human brain vascular atlas reveals diverse mediators of Alzheimer's risk. *Nature* 603, 885-892 (2022).

[2] Finak G, *et al.* "MAST: a flexible statistical framework for assessing transcriptional changes and characterizing heterogeneity in single-cell RNA sequencing data". *Genome Biology* 16, 278 (2015).

Response Figure: Venn diagram of Upregulated and Downregulated DEGs overlapped in different cell types. (A1) Upregulated DEGs overlapped in Astrocyte. (A2) Downregulated DEGs overlapped in Astrocyte. (B1) Upregulated DEGs overlapped in Endothelial. (B2) Downregulated DEGs overlapped in Endothelial. (C1) Upregulated DEGs overlapped in Excitatory neurons. (C2) Downregulated DEGs overlapped in Excitatory neurons. (D1) Upregulated DEGs overlapped in Inhibitory neurons. (D2) Downregulated DEGs overlapped in Inhibitory neurons. (E1) Upregulated DEGs overlapped in Microglia. (E2) Downregulated DEGs overlapped in Microglia. (F1) Upregulated DEGs overlapped in OPC. (F2) Downregulated DEGs overlapped in OPC. (G1) Upregulated DEGs overlapped in Oligodendrocytes. (G2) Downregulated DEGs overlapped in Oligodendrocytes.

7. Developing a free database, reducing barriers and speeding up single-cell research are the great undertaking. The maintenance of a web server is a difficult task. Therefore, the publication of a paper developed a tool is not

the end, but the beginning. However, I noticed that scREAD lost maintenance for a long time after it was released. Have you taken any measures to ensure the maintenance of ssREAD this time?

Response: We appreciate your recognition of the potential value of ssREAD. After publishing scREAD, we started to prepare the new version of scREAD to include more data and analyses. However, due to the inclusion of a large number of spatial transcriptomics data, as well as the integrated analysis of single-cell and spatial data, the backend framework of scREAD is not appropriate for continuous updates. Thus, we decided to introduce ssREAD, instead of directly updating and maintaining scREAD server. We have since implemented several measures to ensure the long-term maintenance and continuous updating of the ssREAD database:

1. Update plan: We plan to update the ssREAD database regularly, with new data and features being added every six months. This will ensure that our users have access to the most recent and relevant data in the field of single-cell research. In the future, we plan to have ssREAD new versions using the same web link.
2. Feedback system: We have established a dedicated feedback channel on the ssREAD website (<https://bmbx.bmi.osumc.edu/ssread/feedback>), where users can report any issues, suggest improvements, or request new features. Our team will review and address the feedback in a timely manner, ensuring that the platform stays up-to-date and responsive to user needs.
3. Server backup system: To ensure the stability and availability of ssREAD, we have set up backup servers and have implemented cloud-based backup solutions (<https://go.osu.edu/ssread>). In case of unavailability of the main server, our development team can switch the link to redirect to an alternative server without requiring users to enter a different URL. To improve transparency and provide real-time updates for our users, we have created a status page for ssREAD (<https://ssread.statuspage.io/>). This page features automated monitoring of the server's status and allows users to track the operational status of the database, including any scheduled maintenance or unexpected downtime.

Issues for the web server.

1. The main functions and analyses that ssREAD performed are comparisons and pathway enrichment. But there are many types of analyses designed for scRNA-seq and ST data, cell communication for example. Further in-depth analysis could be included in ssREAD.

Response: Thank you for your comments. We have included cell-cell communication analysis as an additional feature in ssREAD. This analysis will help users explore potential interactions between different cell populations within the datasets, providing a more comprehensive understanding of the biological processes at play. Our platform currently provides mainstream analysis options for scRNA-seq and ST data, which cater to the majority of users' needs. We are committed to the continuous improvement of ssREAD and plan to add more advanced analysis options in the future. We will actively engage with the research community and stay up-to-date with the latest developments in scRNA-seq and ST data analysis to ensure that our platform remains relevant and useful for researchers.

2. It is nice that the ssREAD provides users variable arguments. But when users change an argument, there is no loading effect to remind users. The reviewer suggest that it should be added.

Response: Thank you for your suggestion. We have improved the server user interface to provide immediate notifications to users when they display data and visualization was updated (**Supplementary Tutorial of ssREAD**). Specifically, we have added loading effects to the following functions: (1) Search for specific gene, (2) find overlapping genes, (3) visualize specific gene in UMAP and violin plot, and (4) perform pathway analysis.

3. For download of the results figures, the text distribution, resolution, and size of the snapshot are not that high quality. This would make it difficult for users to put it directly in their paper.

Response: We appreciate your feedback on the quality of the downloaded figures and understand the importance of providing high-quality images for use in academic papers. We have increased the default resolution of the generated figures to ensure that the images are of sufficient quality for direct use in publications.

This improvement will provide users with clearer and more detailed visualizations. We have also upgraded our visualization framework to offer optional higher-quality SVG vector format figures for download. The vector images can be scaled up or down without losing resolution. This improvement ensures that the resolution, text distribution, and size of the downloaded figures are of sufficient quality to be directly used in academic papers.

4. Additionally, necessary explanation should be added to improve users' sense of experience. For example, if it is not suitable for a sample to perform Disease vs disease (same region) or Disease vs. disease (different region), you should remind the users and give it a reason.

Response: Thank you for your suggestion. Specifically, we have implemented a feature that disables non-biological comparisons from being performed on ssREAD to prevent users from inadvertently conducting analyses that may lead to misleading or scientifically invalid conclusions. We have also enhanced the platform by incorporating reminders and reasons for different situations. (1) When performing DEGs between specific conditions with no appropriate datasets, we show a reminder “Sorry. There are no datasets available for comparison.” (2) When no significant DEGs are identified between selected data, we show “Sorry. No differentially expressed genes are available to present.” (3) When no significantly enriched pathways are identified, we show “Sorry. No pathways have been identified.”

5. DEGS in disease and control is performed between the selected sample and another sample. Whether you performed sample integration to provide real group comparisons? Biological repeatability will improve the reliability of expression difference analysis. Comparing only two samples will lose the advantage of sample size in the original study.

Response: Yes, we integrate biological replications in the same study as a dataset for DEG analysis. For example, AD035 is the ID of the study [1], which involves 84 samples. We grouped the 84 samples into 12 datasets based on sample conditions. The AD03501 dataset includes 2 samples that are both female AD in Braak stage II, the AD03502 dataset includes 3 samples that are all female AD in Braak stage IV, and so on. When selecting “Braak stage II” and “Braak stage IV” groups in female AD, it compares genes between 2 samples in AD03501 and 3 samples in AD03502, rather than individual samples.

[1] Gabitto MI, *et al.* Integrated multimodal cell atlas of Alzheimer's disease. *Res Sq*, (2023).

6. Detailed methods should be provided in the web server. As a web server, the methods description is important for users to replicate the results and use the results in their publications.

Response: Thank you for your comments. We have updated the methods section on the ssREAD website (<https://bmbxl.bmi.osumc.edu/ssread/help/methods>). This update ensures that the descriptions are consistent with the revised manuscript. The enriched sections showcase the analysis of each step, including the dataset collection source, data preprocessing and quality control, data normalization and integration, dimensionality reduction, cell type annotation, differential gene expression analysis, regulon analysis, and cell-cell communication analysis. We have included tables and figures to illustrate the methodology details, such as the marker gene used in the database.

7. After clicking the EXAMPLE button, the functional part under the QUERY module “Find overlapping DEGs from multiple comparisons” is loading all the time and no results are returned.

Response: We appreciate your feedback. We have addressed the loading issue.

8. When searching for differentially expressed genes, the gene name search field should allow the partial match as well, for example, if the user only type ‘Apo’, it should match with ‘ApoE’ gene name.

Response: Thank you for your suggestion. We have added the function to allow the partial match for searching.

9. The speed of the ssREAD is slow. It is better to speed up it to make it more user friendly.

Response: To speed up ssREAD, we have upgraded and optimized our backend framework and frontend visualizations for better memory usage and data loading, especially for processing large datasets.

Reviewer #3

This is a meaningful work, and research on Alzheimer's Disease indeed requires continuously updated databases to support it, especially with the development of spatial transcriptomics and single-cell technologies, as well as their applications in neuroscience.

After reading the article:

1. The article mentions the comparison between ssREAD and other similar databases, and I checked the table including Single-cell portal and STOmicsDB, and found that the numbers may not be accurate, which could be due to the information being updated until May 2023, and it may need to be confirmed again;

Response: We appreciate your comments and recognition of the importance of continuously updated databases in AD research. We have re-evaluated and updated the comparison between ssREAD and other similar databases, such as the Single-cell portal and STOmicsDB, to ensure accuracy. The revised table (**Supplementary Table S1**) reflects the most recent information available by December 2023.

2. I compared ssREAD with scREAD and found that ssREAD has made many improvements in UI/UX, but the article also mentions some analyses done after incorporating spatial transcriptomics data. It would be helpful to provide some online analysis tools on the website, as currently it mainly focuses on static page display and visualization.

Response: We appreciate your suggestion. The primary focus of ssREAD is to build a comprehensive and user-friendly database, rather than an online analysis web server. However, we understand the value of providing interactive analysis tools to enhance the user experience and facilitate exploring spatial transcriptomics data. We have implemented several interactive analysis features on the ssREAD website as long as more stable maintenance features, such as:

1. Users can upload their own data and request specific data analysis on ssREAD. We will evaluate the request and run the analysis. Eventually, the user will receive a specific link with their data analysis results.
2. Users can now perform pathway analysis based on the genes of interest, obtaining results in real time.
3. Users can search for specific genes and visualize their expression patterns across different datasets.
4. The platform allows users to find overlapping genes between different datasets, aiding in discovering common gene signatures.
5. Users can tailor the visualization of specific genes or pathways according to their research needs.

We believe that these updates strike a balance between providing a comprehensive database and offering interactive analysis tools that cater to the needs of researchers working with spatial transcriptomics and single-cell data in AD research.

REVIEWERS' COMMENTS

Reviewer #2 (Remarks to the Author):

The author had addressed all of my concerns.
The reviewer think that this manuscript is ready for publication.

Reviewer #3 (Remarks to the Author):

Very grateful for the database author's rapid iterations and optimizations. When using the upgraded database, I noticed a few minor issues:

On the page <https://bmbxl.bmi.osumc.edu/ssread/singlecell>, in the table under "scRNA-seq datasets", there may be some bugs:

1. For example, when clicking on the "GEO/synapse ID" column, such as clicking on syn18485175 or syn21125841, the new page I'm redirected to contains errors in the backend code or database links assembled in your code. For example, syn18485175; syn21125841 should have separate links for each ID.

2. Also, when I scroll to copy these IDs, a popup always appears. Or when I click on the ID link to open a new page and then go back to the original <https://bmbxl.bmi.osumc.edu/ssread/singlecell> page, a popup also appears. But I'm not sure if this is due to some software on my Mac, so you may want to confirm.

Additionally, I'm using Chrome Version 120.0.6099.234 (Official Build) (x86_64). This information may help you test.

And at the bottom of the <https://bmbxl.bmi.osumc.edu/ssread/downloads> page is the download table for Spatial Transcriptomics datasets. It would be better if descriptive information corresponding to the ssREAD IDs could be added.

Reviewer #3

On the page <https://bmblix.bmi.osumc.edu/ssread/singlecell>, in the table under "scRNA-seq datasets", there may be some bugs:

1. For example, when clicking on the "GEO/synapse ID" column, such as clicking on syn18485175 or syn21125841, the new page I'm redirected to contains errors in the backend code or database links assembled in your code. For example, syn18485175; syn21125841 should have separate links for each ID.

Response: We appreciate your observation regarding the "GEO/synapse ID" column on the table page. We have amended our code to ensure that each ID now has a separate, functional link.

2. Also, when I scroll to copy these IDs, a popup always appears. Or when I click on the ID link to open a new page and then go back to the original <https://bmblix.bmi.osumc.edu/ssread/singlecell> page, a popup also appears. But I'm not sure if this is due to some software on my Mac, so you may want to confirm. Additionally, I'm using Chrome Version 120.0.6099.234 (Official Build) (x86_64). This information may help you test.

Response: Thank you for the testing. we have conducted tests on various platforms, including the Mac version of Chrome that you mentioned. We have confirmed that the issue was not specific to your Mac software, and we have now resolved the problem. We have added an individual button to open the pop-up window to ensure a smoother user experience.

3. And at the bottom of the <https://bmblix.bmi.osumc.edu/ssread/downloads> page is the download table for Spatial Transcriptomics datasets. It would be better if descriptive information corresponding to the ssREAD IDs could be added.

Response: We appreciate your comments. We have enhanced the download table with corresponding descriptive information for each ssREAD ID. Furthermore, we have provided references for each dataset available for download on the page.